# Non-invasive estimation of haemodynamic parameters in pulmonary hypertension — A deep learning approach integrating all B-mode cine loops in an echocardiographic exam

**Li-Hsin Cheng**[1]  ID                                L.CHENG@LUMC.NL

**Samer Alabed**[2]                           S.ALABED@SHEFFIELD.AC.UK

**Athanasios Charalampopoulos**[3]    ATHANASIOS.CHARALAMPOPOULOS@NHS.NET

**Ze Ming Goh**[2]                          Z.M.GOH@SHEFFIELD.AC.UK

**Abdul Hameed**[2,3]                         AHAMEED@NHS.NET

**Eduard Holman**[1]                       E.R.HOLMAN@LUMC.NL

**David G. Kiely**[2,3,4]                    DAVID.KIELY1@NHS.NET

**Mahan Salehi**[2]                        M.SALEHI@SHEFFIELD.AC.UK

**Andrew J. Swift**[2]                    A.J.SWIFT@SHEFFIELD.AC.UK

**Rob J. van der Geest**[1]           R.J.VAN_DER_GEEST@LUMC.NL

[1] *Leiden University Medical Center, the Netherlands*

[2] *University of Sheffield, UK*

[3] *Sheffield Teaching Hospitals NHS Foundation Trust, UK*

[4] *National Institute for Health and Care Research Sheffield Biomedical Research Centre, UK*

**Editors:** Accepted for publication at MIDL 2026

## Abstract

Pulmonary hypertension (PH) is heterogeneous with treatment strategy dependent on the underlying cause and disease severity. Haemodynamic parameters measured through right heart catheterization (RHC) is the gold standard for such diagnosis and desicion making. However, the invasive procedure is associated with a certain level of risk and is not suitable for every patient. Therefore, we seek to investigate whether haemodynamic parameters can be estimated non-invasively using a deep learning approach. The study is based on a retrospective analysis of 833 subjects with suspected PH identified from the ASPIRE research database. Convolutional neural networks were built to integrate B-mode echocardiographic cine loops from multiple views to predict key haemodynamic parameters. The model was able to integrate an arbitrary number of cine loops in the entire exam, unannotated with view names. Additionally, attention weights in feature fusion identify relevant and irrelevant cine loops to the model. The model-predicted mean pulmonary artery pressure (mPAP) correlated to the RHC-ground truth with a Pearson Correlation Coefficient (PCC) of 0.70. Attention weights indicated the apical 4-chamber (A4C) view to be especially relevant for mPAP prediction. Our results demonstrate the feasibility of estimating haemodynamic parameters non-invasively through deep learning models, integrating all B-mode cine loops of a cardiac ultrasound exam, achieving a moderate correlation to RHC measurements.

**Keywords:** Pulmonary hypertension, mean pulmonary artery pressure, multi-view integration, B-mode echocardiography, deep learning.

## 1. Introduction

Pulmonary hypertension (PH) is heterogeneous with treatment strategy dependent on the underlying cause and disease severity (Kondo et al., 2019). For patients suspected of having PH, echocardiography is recommended as the first line investigation (Humbert et al., 2022). A subsequent right heart catheterization (RHC) is recommended to confirm the diagnosis. According to the 2022 ESC/ERS Guidelines (Humbert et al., 2022), PH is defined as the mean pulmonary artery pressure (mPAP) measured by RHC being higher than 20 mmHg. Together with pulmonary vascular resistance (PVR) and pulmonary arterial wedge pressure (PAWP), the haemodynamic profile obtained from RHC is an important indication about the cause of elevated pulmonary artery pressure and the severity. By summarizing a range of investigations, patients are categorized into one of five main PH groups to permit appropriate clinical management.

The haemodynamic parameters are critical indicators. However, the invasive RHC procedure to obtain the parameters is associated with a certain level of risk, and may not be suitable for every patient (Rosenkranz and Preston, 2015; Hoeper et al., 2006). Consequently, a non-invasive alternative to RHC would be clinically valuable. Echocardiography is an ideal modality for deriving such non-invasive estimates, as it is already part of the clinical routine. Cardiac ultrasound offers unique advantages, including being portable, low cost, safe (without radiation), and fast to operate. Furthermore, cardiac ultrasound allows for real-time imaging of the heart from multiple angles (views). Therefore, this study explores a non-invasive approach to estimate haemodynamic parameters from echocardiography.

Currently, from echocardiography, the key quantitative parameter that can be obtained to assess the probability of PH is the maximal tricuspid regurgitation velocity (TRVmax) (Humbert et al., 2022) as derived from continuous wave Doppler. A TRVmax higher than 2.8 m/s may suggest PH. However, that alone is not yet sufficient to reliably determine the presence or absence of PH. More importantly, the approach is limited to only patients with a measurable tricuspid regurgitation (Mazurek and Forfia, 2013). In terms of visual qualitative assessment with echocardiography, there are various signs for inferring the probability of PH, for example, enlargement of the right ventricle (RV) in the parasternal long-axis (PLAX) view, flattened interventricular septum in the parasternal short-axis (PSAX) view, or enlarged right atrium (RA) or RV in the apical 4-chamber (A4C) view (Humbert et al., 2022; Kiely et al., 2019). To derive estimations independent of tricuspid regurgitation and to integrate comprehensive information from multiple views, utilizing all the B-mode images instead of relying solely on TRVmax is a logical approach. However, there are currently no recommendations to systematically summarize findings across all B-mode views to conclude PH.

Deep learning is a promising method that allows summarizing massive inputs quantitatively and objectively. There have been several related studies (Diller et al., 2022; Ragnarsdottir et al., 2024; Zhang et al., 2018) which leverage deep learning methods to classify various PH-related categories using selected B-mode views. Machine learning approaches (Hirata et al., 2024; Vaidya et al., 2020; Leha et al., 2019; Liao et al., 2023; Anand et al., 2024; Swinnen et al., 2023), on the other hand, rely on predefined echo-derived measurements as inputs, which offers model transparancy but are limited in their ability to directly process massive image inputs. We based our study on the success of previous deep learn-

ing approaches and identified further challenges to overcome in terms of the model design. First, a complete haemodynamic profile is essential for determining the underlying cause and severity of PH, thereby guiding appropriate clinical management. Therefore, ideally the model should estimate all key haemodynamic parameters as continuous values, instead of making coarse classification. Second, in real-world echocardiographic data, the view of a given scan is unknown unless explicitly annotated. Therefore, ideally the model should be capable of accepting view-unannotated cine loops as input. Third, predefining a set of input views without prospectively verifying their optimality prevent the discovery of unexpected highly diagnostic views, thus potentially limiting the predictive performance. Therefore, it would be beneficial to provide the model access to all B-mode views available in an exam. Finally, the diversity and the noise in echocardiographic dataset pose challenges to image analysis. In terms of diversity, there are more views in echocardiography than in other modalities. Within each major view category, the images are also less standardized due to different zoom levels, focal points, or variances introduced by slightly different acquisition angles. In terms of noise, ultrasound images are inherently affected by speckle noise and suboptimal acquisition which stay in the dataset. Being able to make accurate predictions from highly diverse and noisy images is therefore an important consideration in model design.

In summary, the study aims to determine whether haemodynamic parameters can be estimated non-invasively based on echocardiographic images using a deep learning approach. To achieve this goal, a sub-focus is to integrate all B-mode cine loops within an exam, ensuring a more comprehensive analysis that remains effective even in the absence of tricuspid regurgitation. Another sub-focus concerns deep learning model design that: (1) predicts continuous haemodynamic parameters directly instead of making coarse classification, (2) handles an arbitrary number of view-unannotated cine loops in each exam, (3) does not restrict the input views to the model, and finally, (4) can navigate through the noisy, diverse images.

## 2. Methods

### 2.1. Patient cohort and metadata

In this retrospective study, patients with suspected PH who were referred to Sheffield Pulmonary Vascular Disease Unit from 2009 and 2021 were identified from the ASPIRE database (Assessing the Spectrum of Pulmonary Hypertension Identified at a Referral Center) (Salehi et al., 2025; Hurdman et al., 2012). All included patients underwent both echocardiography and RHC within 3 months. Echocardiography was acquired with GE Vivid machines. RHC was performed as part of the routine clinical pathway, using a balloon-tip 7.5-F thermodilution catheter (Becton-Dickinson, Franklin Lakes, NJ). In 59% of the cases, echocardiography was obtained prior to RHC. In 40% of the cases, RHC was obtained first. In 1% of the cases, the two were obtained on the same day. The 1014 patients were randomly split into training, validation, and testing sets with the ratio of 65%, 15%, 20%. After exclusion of invalid exams, a total of 833 patients remained for analysis. Details about the exclusion can be found in Appendix A. Patient characteristics are summarized in Table 1.

Table 1: Patient characteristics.

|  | Training | Validation | Testing |
|---|---|---|---|
| **Number of subjects** | 545 | 129 | 159 |
| Sex female (n, %) | 378, 69% | 94, 71% | 100, 62% |
| **Number of exams** | 550 | 132 | 160 |
| Age (years) | 63.9±14.0 | 62.6±13.4 | 63.8±15.1 |
| WHO functional class (%)[†] | 1/13/72/10/4 | 2/9/73/13/3 | 1/9/74/10/6 |
| Echo–RHC time difference (days)[‡] | 21.2±15.9 | 20.7±14.6 | 19.1±15.8 |
| mPAP (mmHg) | 36.7±16.2 | 37.5±16.8 | 36.9±15.1 |
| PAWP (mmHg) | 11.9±5.8 | 11.6±5.6 | 11.9±5.6 |
| PVR (Wood units) | 6.2±5.2 | 6.1±4.8 | 5.9±4.5 |

[†]WHO functional classes: I/II/III/IV/missing. [‡]Absolute time difference between two exams.

## 2.2. Echocardiographic images

In this study, B-mode echocardiography cine loops covering at least one heartbeat were used. The view (acquisition angle) of each cine loop is not annotated. The number of B-mode cine loops per exam, duration of each cine loop, and width and height of the scanning sector are listed in Appendix B.

The scanning sector was cropped from the raw DICOM image according to the bounding box information provided in the DICOM header. The pixel signal intensity, which is stored as RGB-values in the original DICOM images, was converted into grayscale and scaled to a range from 0 to 1. The image was then center-cropped or zero-padded to a square of the image height, then resized to 256 pixels × 256 pixels. During training, additional random rotation (± 15°), scaling (± 10%), and shifting (± 10%) was applied.

## 2.3. Multi-view deep learning model

We constructed the model based on the multi-view cine series deep learning model as described previously (Cheng et al., 2025). This model works on four fixed MR views, whereas the adapted model accepts all B-mode cine loops from an echocardiographic exam as input, and predicts the associated mPAP. An overview of the model is presented in Figure 1(a). First, the Frame Encoder encodes each frame into a frame feature. The encoder has a ResNet50 (He et al., 2016) architecture. Next, the frame features are fused into the final feature through multiple steps. Finally, the Regression Layer, which is a single linear layer, transforms the final feature into a mPAP prediction.

The fusions (Figure 1(b)) are facillitated by the attention mechanism, specifically, the Attention Feature Fusion Block (AFFB) (Cheng et al., 2025). The AFFB assigns a weight to each input component feature, and the output fused feature is formed accordingly by a weighted sum of the component features. The weights are generated by a single linear layer within the AFFB. The higher the assigned weight, the higher proportion a component makes up the fused feature. The attention weights thus provide an indication which features are considered relevant or not by the model. In temporal frame fusion (Figure 1(b)), frame features from a cine loop (gray circles) are fused by the AFFB into a view summary (gray

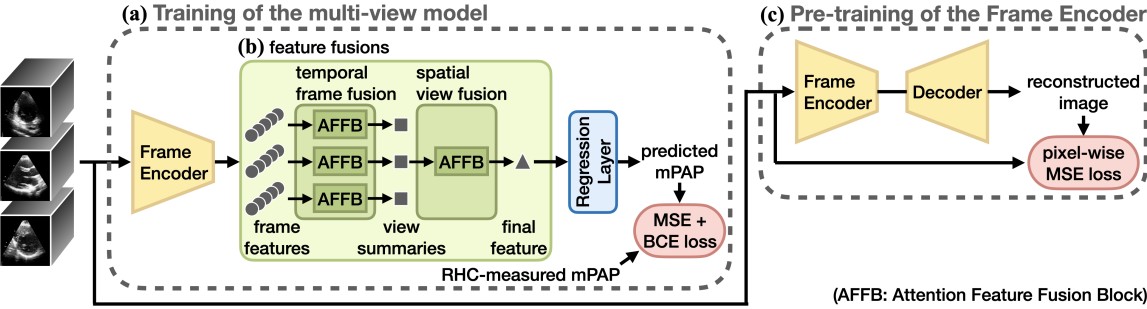

Figure 1: Model overview. (a) The multi-view deep learning model accepts an arbitrary number of B-mode cine loops as input and predicts mPAP. (b) The frame features are fused into view summaries, then into the final feature, facilitated by the AFFB. (c) The Frame Encoder is first pre-trained with an image reconstruction task, which learns the representation of a diversity of echocardiographic images.

squares). Each view summary then represents one B-mode cine loop from the exam. In spatial view fusion, multiple view summaries are fused by the AFFB into the final feature (gray triangle), representing the final aggregation of all B-mode images in the entire exam. We inspected especially the attention weights in spatial view fusion to analyze relevant cine loops (Section 2.4).

Additionally, we pre-trained the Frame Encoder (Figure 1(c)), to familiarize it with the variety in echocardiographic images, including a diversity of views, zooms, and image qualities. The pre-training is an image reconstruction task, where the autoencoder without skip connections learns to reconstruct any given input image. The decoder for reconstruction is thrown away after pre-training, while the encoder has learnt to map images into informative feature embeddings and was used as the initialization for the actual training of the multi-view model.

The study focusses mainly on mPAP prediction, but an identical approach was used to build two additional models for PVR and PAWP predictions.

## 2.4. Analyzing relevant cine loops for mPAP prediction

We aimed to find out the characteristics of cine loops which contributed highly to mPAP predictions, and the characteristics of cine loops which were suppressed by the model. For this purpose, we inspected the attention weights assigned to each view summaries (gray squares in Figure 1(b)) during the spatial view fusion. A view summary represents a cine loop. During spatial view fusion, a view summary that was assigned a higher attention weight made up a higher portion of the final feature, dominating the final mPAP prediction. The corresponding cine loop was thus considered relevant. Similarly, a low weight assigned to a view summary implied that the corresponding cine loop was considered irrelevant by the model.

Representative highly relevant cine loops for analyses were isolated as following. For each exam, the highest weighted view summary (cine loop) was identified. Since visual analysis of the cine loops from each of the 160 testing exams would be too demanding,

among these 160 representatives, the top 10% ($n = 16$) with the highest raw attention weights were further isolated as the representative of highly relevant cine loops. Similarly, representative irrelevant cine loops were obtained by identifying the least weighted view summary in each exam ($n = 160$), then further isolating the bottom 10% ($n = 16$) with the lowest raw attention weights.

We assessed the commonality among the 16 representative relevant cine loops, and the commonality among the 16 representative irrelevant cine loops. The view name and image quality of each identified cine loop were manually annotated by cardiac ultrasound experts (AH and AC respectively), and the analysis results are shown in Section 3.3.

## 2.5. Implementation details

An NVIDIA RTX6000 GPU with 24 GB of memory was used for building the models. The models were all implemented with Pytorch and Pytorch Lightning. The Adam optimizer was used, with the learning rate set to $10^{-4}$ and weight decay set to $10^{-8}$. The batch size was set to 12.

During the pre-training of the Frame Encoder, the image-reconstruction autoencoder was trained for 800 epochs with early stopping. The loss was pixel-wise Mean Square Error (MSE) loss between the input and the reconstructed image. The testing data was not used at this stage, in order to avoid leaking the information. Specifically, a proportion of the training data was used for training the autoencoder and the rest of the training data was isolated for early stopping. The validation data was used to evaluate the autoencoder.

For the actual training of the multi-view model, the entire model was trained for 100 epochs with early stopping. The training, validation, and testing data were used to train, early stop, and evaluate the model, respectively. The predictive performances presented in the entire Results Section are based on the testing data. The loss was the average of a regression loss and a classification loss. The regression loss component was the MSE between the predicted and the RHC-measured ground-truth mPAP. The classification loss component was a binary cross entropy (BCE) loss between the predicted and the RHC-measured mPAP, both being thresholded at 20 mmHg. It encourages careful delineation around the clinically important threshold under a regression scheme. For efficiency, during each training iteration, 5 cine loops were randomly drawn from each exam, and 5 random frames were drawn from each cine loop. For a comprehensive ensembling, during evaluation, all the available B-mode cine loops in an exam were included as the model input, each consisting of 10 frames sampled uniformly from the first heartbeat.

## 3. Results

### 3.1. mPAP estimated through B-mode cine loops from multiple views

TRVmax is the existing way to estimate PH probability non-invasively. For the testing exams which TRVmax was available ($n = 122$), the TRVmax correlates to the RHC-measured ground-truth mPAP with Pearson correlation coefficient (PCC) of 0.72. On the other hand, the model's mPAP prediction correlates to the ground truth with a PCC of 0.70, deviating on average 8.77 mmHg from the ground truth (Table 2).

Table 2: Regression performance. TRVmax and model predictions of mPAP, PVR, PAWP evaluated against respective RHC-measured ground truths.

|  | **TRVmax** (n=122) | **mPAP** (n=160) | **PVR** (n=160) | **PAWP** (n=160) |
|---|---|---|---|---|
| **MAE**[†] | - | 8.77 (mmHg) | 2.57 (WU) | 3.65 (mmHg) |
| **PCC** | 0.72 (p<0.05) | 0.70 (p<0.05) | 0.68 (p<0.05) | 0.54 (p<0.05) |

[†]MAE: Mean Absolute Error.

Using the common 2.8 m/s threshold, TRVmax misclassified 11 non-PH cases as PH. TRVmax classified PH with an accuracy of 0.84, sensitivity of 0.92, specificity of 0.54, and area under receiver operating characteristic curve (AUC) of 0.88 (Figure 2(a,e)).

The deep learning model tends to predict mPAP near the mean while struggling with extreme values (Figure 2(b)), mis-classifying 28 non-PH cases as PH (Figure 2(f)). When using the common 20 mmHg threshold, the model classified PH with an accuracy of 0.78, sensitivity of 0.94, specificity of 0.24, and AUC of 0.83.

### 3.2. Non-invasive haemodynamic profiling

Using the same approach as the mPAP model, PVR and PAWP prediction models were built. Results indicate that PVR can be estimated non-invasively using echocardiographic images with an average error of 2.57 Woods units (WU), and PCC of 0.68. The predicted

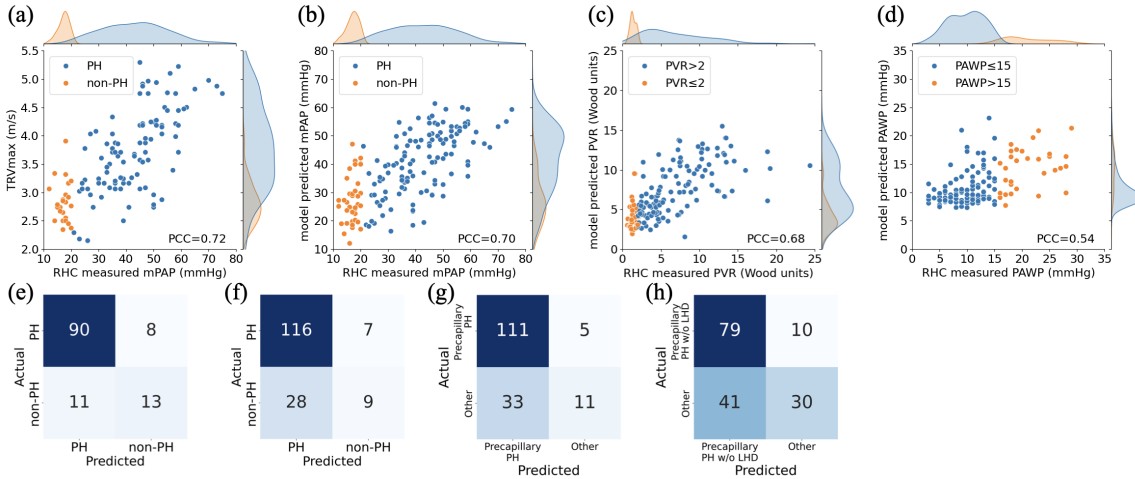

Figure 2: Model predictions. Scatter plots of (a) TRVmax, and (b-d) predicted mPAP, PVR, PAWP, against respective RHC-ground truth. On the side of each axis is a kernel density estimation depicting the distribution of the values. Confusion matrices of (e) PH classification through TRVmax thresholded at 2.8 m/s, (f) PH classification through model prediction, (g) pre-capillary PH classification, (h) pre-capillary PH without LHD classification.

PAWP correlates with the ground truth only moderately but statistically significantly. The average error was 3.65 mmHg and the PCC was 0.54 (Table 2). The predicted PVR correlates well with the ground truth in general, but is less accurate when PVR $\leq$ 2 WU (Figure 2(c)). The prediction error was higher in cases with higher PAWP (Figure 2(d)).

Combining the mPAP, PVR, and PAWP models together enables the prediction of the haemodynamic profile from echocardiographic images. The models identified pre-capillary PH (mPAP > 20 mmHg, PVR > 2 WU) with an accuracy of 0.76, sensitivity of 0.96, and specificity of 0.25 (Figure 2(g)). Due to higher errors in identifying mPAP $\leq$ 20 mmHg and PVR $\leq$ 2 WU, the false positive rate for detection of pre-capillary PH was also higher. The models predicted pre-capillary PH in the absence of left heart disease (LHD) (mPAP > 20 mmHg, PVR > 2 WU, PAWP $\leq$ 15 mmHg) with an accuracy of 0.68, sensitivity of 0.89, and specificity of 0.42 (Figure 2(h)). The lower accuracy in this classification was primarily due to challenges in PAWP prediction.

### 3.3. Relevant view and image quality for mPAP prediction

We analyzed common characteristics among the cine loops that are the highest, or the least weighted by the model to form mPAP predictions, respectively, in terms of view and image quality.

Figure 3(Left) presents the cine loops that receive especially high attention weights. The model considers them to be relevant and uses a high proportion of the derived features in forming the mPAP prediction for the respective exams. The model-preferred cine loops was most often the A4C view (12 out of 16), and most often graded as acceptable image quality (11 out of 16). Overall, the model has a strong preference in forming the final feature (and thus the final mPAP prediction) with a high proportion of an A4C cine loop that has an image quality of acceptable (9 out of 16).

Figure 3(Right) presents the cine loops that receive especially low attention weights. The model considers them to be irrelevant and uses a low proportion of the derived features in forming the mPAP prediction for the respective exams. The model's least preferred cine loops were most often (11 out of 16) graded as suboptimal image quality, but there appears no strong consensus in the view. Four out of 16 were from the subcostal inferior vena cava (IVC) view, while three were from the subcostal four chamber (4CH) view. This indicates that the model primarily disregards cine loops with suboptimal image quality, rather than specific views.

## 4. Discussion

This study demonstrates the potential of estimating haemodynamic parameters non-invasively through deep learning models, integrating all B-mode cine loops from multiple views in an echocardiographic exam. This approach offers a possibility to avoid the risk associated with RHC, and to summarize the complex echo data quantitatively and objectively.

Non-invasive haemodynamic estimation can be derived from several imaging modalities. This study focused on echocardiography, and specifically B-mode images, achieving a PCC of 0.70 for mPAP prediction. Our model classified PH with an accuracy of 0.78, a sensitivity of 0.94, and a specificity of 0.24. In a previous study, Cheng et al. (2025) achieved a PCC of 0.80 for mPAP prediction using cardiac MRI as input. The superior image quality of

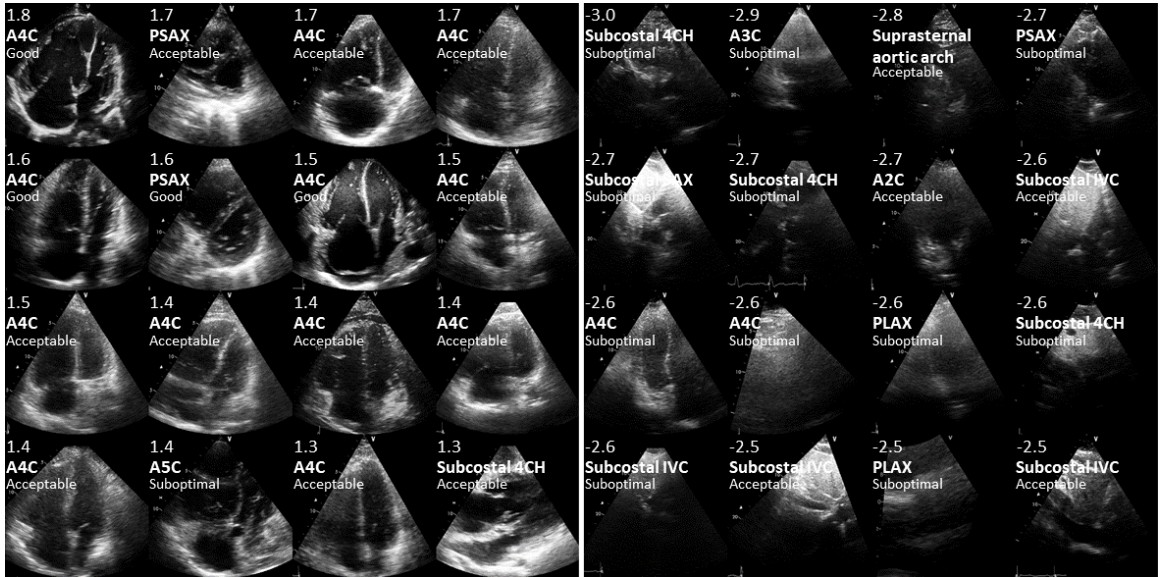

Figure 3: Representative cine loops regarding mPAP prediction. (Left): relevant cine loops. (Right): irrelevant cine loops. At the top-left of each cine loop are the raw attention weight, and the view name and image quality as annotated by the echo expert.

MR or the larger dataset might have contributed to its better performance. Nevertheless, echocardiography offers unique advantages of being a standard initial diagnostic investigation performed routinely in clinical practice. Additional advantages include portability, low cost, safe, relative ease in performing, time required, and real-time imaging. Therefore, echocardiography remains a strong modality to serve scenarios that relate to a screening or assistive purpose. To derive quantifications similarly from the echocardiography modality, but using Doppler measured TRVmax instead of using B-mode images, the PCC (0.72) is similar while the specificity (0.54) is higher. However, TRVmax is limited to patients having a measurable tricuspid regurgitation signal. In fact, 38 exams in the testing set (24% of the testing set) had no documented TRVmax, but our method managed to estimate all three haemodynamic parameters from these exams. Additionally, B-mode images provide a more comprehensive view of heart morphology and motion, presenting a greater potential for improvements in the future.

There are other related works which adopted both B-mode images and deep learning methods. However, these studies all focus on classification targets. For example, distinguishing between pulmonary arterial hypertension (PAH), RV dilation without PAH, and absence of cardiac abnormality (Diller et al., 2022); distinguishing non-PH, mild PH, and severe PH (Ragnarsdottir et al., 2024); or distinguishing PAH from healthy control (Zhang et al., 2018). We argue that estimating all important haemodynamic parameters as continuous values is a more fundamental framework. It covers all possible variations in haemodynamic profile and severity. Moreover, these prior studies relied on datasets with view annotations, and developed the model directly on selected images. However, this is usually not a given,

and our model offers the possibility to utilize a view-unannotated dataset. Finally, previous studies preselected input views without verifying their optimality, potentially limiting predictive performance. For example, Diller et al. (2022) used A4C and parasternal 2-chamber view, in combination with echo-estimated RV systolic pressure (RVSP). Ragnarsdottir et al. (2024) used PLAX, A4C, and PSAX views at three levels. Zhang et al. (2018) used the A4C view. In contrast, our framework allows the model to analyze all B-mode images, then select predictive features without human bias.

The attention mechanism provides additional advantages in handling noisy and diverse datasets. In terms of navigating through a noisy dataset, the model assigned lower attention weights to cine loops with poor image quality, reducing their influence on the final predictions. In terms of handling the high diversity of the dataset, the model identified the A4C view as especially relevant for mPAP prediction, which aligns with clinical guidelines (Humbert et al., 2022). Although the model did not reveal novel clinical features, technically, it is still notable for three reasons: First, the model can work without enforcing prior assumptions. Second, it confirmed current clinical knowledge through an independent data-driven approach. Third, it provided a reference for future studies, especially if the focus shifts to minimizing the number of required views while maintaining performance.

Our proposed method is non-invasive, automated, and capable of predicting the parameters in continuous values. It could support clinical workflows in several ways. Early Assessment: providing quantitative estimates already at the initial evaluation, even in cases without measurable tricuspid regurgitation. Guiding RHC Decisions: assisting clinicians in determining the necessity of RHC, based on the model-predicted haemodynamic profile. Monitoring Disease Progression: tracking changes over time to assess disease progression or treatment response, reducing the need for repeated invasive procedures. Triage: prioritizing cases for expert review when clinical workload is high.

A major limitation of the current work concerns the predictive performance. Particularly, the specificity of the mPAP model remains to be improved. The high false positive rate likely arises from dataset imbalance, as the cohort contains more patients with high mPAP than those with normal values. Addressing this may require expanding the dataset to include more non-PH cases or employing data-balancing techniques. Another limitation arises from the potential variations in patient status between the RHC and echocardiography. It would be ideal if the two tests would be performed closer to each other, to further reduce potential label noise. Future work should also explore integrating all the available images from an echocardiographic exam, for example, Doppler or M-mode images. The main challenge will be developing preprocessing and encoding strategies for effective feature fusion. Finally, model interpretability remains limited. If view annotations become available in the complete testing set, analyzing individual view contributions could refine our understanding of their importance. Further investigations could also identify critical image sub-regions, cardiac phases, or anatomical structures most relevant for prediction.

## 5. Conclusion

Our results demonstrated the feasibility of estimating haemodynamic parameters non-invasively through deep learning models, integrating all B-mode cine loops of a cardiac ultrasound exam, achieving a moderate correlation to the RHC measurements.

## Acknowledgments

This study is partly funded by the National Institute for Health and Care Research (NIHR) Sheffield Biomedical Research Centre (NIHR203321). The research was also supported by NIHR AI in Health and Care award (grant no. AI_AWARD01706). The views expressed are those of the authors and not necessarily those of the NIHR or the Department of Health and Social Care.

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

## Appendix A. Data filtering

From the initial 1014 subjects, we excluded (i) invalid exams, (ii) invalid image files, then finally (iii) exams with missing RHC values. The derived final cohort consists of 833 subjects (Table 3). After the exclusion, some subjects have no exams left anymore and thus the resulting final cohort has less subjects than the initial raw cohort.

In step (i), "invalid exams" includes those that have irregular name, missing folder, or are acquired other than GE machines. The step (ii), the exclusion of "invalid image files" is implemented as Table 4, to filter out only B-mode cine loops covering at least 1 heartbeat from all possible image types. Under the "secondary" folder in each ultrasound series lies the DICOM files which are potentially B-mode cine loops. The DICOM files that do not satisfy the constrains listed in Table 4 are invalid and thus excluded.

Table 3: Data filtering.

|  | # subjects | # exams |
|---|---|---|
| **Initial raw cohort** | 1014 | 1031 |
| → (i) Exclude invalid exams | | |
| → (ii) Exclude invalid image files | | |
| **Image-valid cohort** | 933 | 943 |
| → (iii) Exclude exams missing RHC values | | |
| **Final cohort** | 833 | 842 |

Table 4: Constrains to exclude invalid image files and filter out only B-mode cine loops covering at least 1 heartbeat.

| Purpose | Implementation |
|---|---|
| Contains essential tags | SequenceOfUltrasoundRegions, RegionSpatialFormat, RegionDataType, RWaveTimeVector, and FrameTime are not missing in the DICOM header |
| Contains only 1 imaging sector | The number of SequenceOfUltrasoundRegions tags in the DICOM header is exactly 1 |
| Is 2D imaging | The value of RegionSpatialFormat in the DICOM header is exactly 1 |
| Is tissue imaging | The value of RegionDataType in the DICOM header is exactly 1 |
| Covers at least 1 heart beat | RWaveTimeVector in the DICOM header contains at least 2 elements |
| Is a video | The number of dimensions in PixelArray is exactly 4 |

## Appendix B. Echocardiographic images

Per exam, the number of valid B-mode cine loops is presented in Figure 4(a). Per B-mode cine loop, the spatial coverage is presented in Figure 4(b) and the temporal coverage is presented in Figure 4(c). The image preprocessing is illustrated in Figure 5.

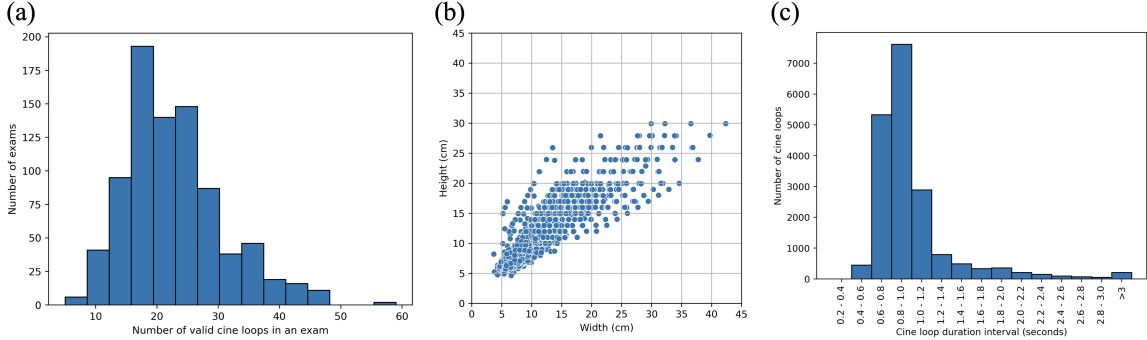

Figure 4: Coverage of the final cohort. (a) The number of valid B-mode cine loops in an exam. (b) The width and height of the scanning sector in each valid B-mode cine loop. (c) The duration of each valid B-mode cine loop.

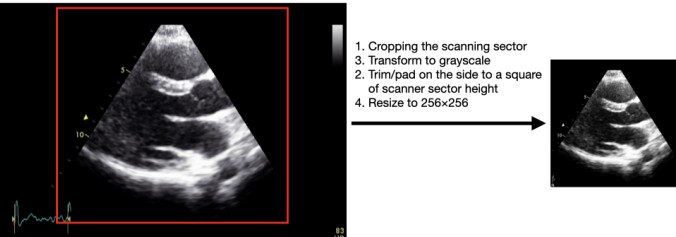

Figure 5: An example of the preprocessed image. From the raw DICOM image, the scanning sector was cropped then converted into a grayscale square image.

## Appendix C. Ablation study

Model settings and hyperparameters were decided based on the ablation study for mPAP prediction. In the ablation study, the final testing data was not involved — A subset from the training set was isolated for early stopping, and the performance on the validation set (as in in Table 5) was used to compare each model setting. After concluding a model setting based on the ablation study, the entire training set was used to train the final model, using the validation set for early stopping, and the testing set to test and derive the final results presented in the main text. Setting (a) was the final setting concluded from the ablation study. The PCC was lower when the Frame Encoder was not pre-trained. Namely, the entire model was trained from scratch for the regression objective (setting (b)). If the

BCE loss was not added to the total loss (setting (c)), the specificity was by default lower. This was probably because the model did not have a special focus on the samples with mPAP lower than 20 mmHg, and aimed only for minimizing the overall MSE. Furthermore, basic class-balancing techniques were experimented. In setting (d), instead of BCE loss, a weighted BCE loss was used, which down-weights the loss for PH exams. In setting (e), weighted sampling was used, where the non-PH exams were oversampled during mini-batch formation. Both class-balancing attempts (settings (d, e)) did not further improve the specificity, and thus were not adopted in the final model.

Table 5: Comparison of different model settings. The ablation evaluation was done on the validation set. Pre-train: to pre-train the Frame Encoder. BCE loss: to include BCE loss as part of the loss function. Weighted BCE: to down-weight the loss for the majority class. Weighted sampling: to draw exams from the minority class more frequently when forming mini-batch. (Acc: accuracy. Sens: sensitivity. Spec: specificity.)

|  | Pre-train | BCE loss | Weighted BCE | Weighted sampling | MAE (mmHg) | PCC | Acc. | Sens. | Spec. | AUC |
|---|---|---|---|---|---|---|---|---|---|---|
| (a) | ✓ | ✓ | × | × | 8.74 | 0.73 | 0.78 | 0.97 | 0.30 | 0.83 |
| (b) | × | ✓ | × | × | 10.05 | 0.66 | 0.75 | 0.98 | 0.16 | 0.84 |
| (c) | ✓ | × | × | × | 9.31 | 0.72 | 0.76 | 0.98 | 0.19 | 0.82 |
| (d) | ✓ | ✓ | ✓ | × | 9.35 | 0.71 | 0.78 | 0.98 | 0.27 | 0.83 |
| (e) | ✓ | ✓ | × | ✓ | 9.48 | 0.69 | 0.75 | 0.93 | 0.30 | 0.82 |

## Appendix D. mPAP prediction details

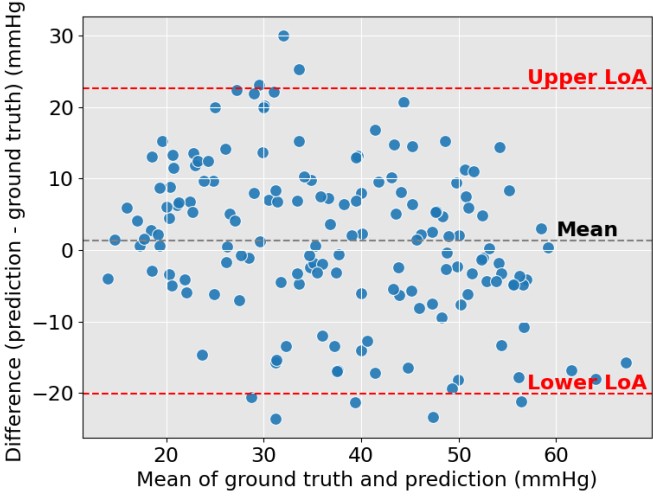

Figure 6: Bland–Altman plot of mPAP prediction.

The Bland–Altman analysis (Figure 6) showed a positive mean bias (1.3 mmHg), and a wide limits of agreement (-20.0 to 22.6 mmHg), indicating substantial variability at the individual level. The predictive performance was further calculated for each subgroup (Table 6), indicating that the largest error was still observed in non-PH subjects, which remains the major point to improve in future works.

Table 6: mPAP predictive performance across subgroups. (CpcPH: combined post- & pre-capillary PH. IpcPH: Isolated post-capillary PH.)

| Subgroup | | Count (# of exams) | MAE (mmHg) | PCC |
|---|---|---|---|---|
| Non-PH | mPAP $\leq$ 20 | 37 | 10.69 | 0.28 |
| PH | mPAP $>$ 20 | 123 | 8.19 | 0.61 |
| Pre-capillary PH | PVR $>$ 2, PAWP $\leq$ 15 | 89 | 8.06 | 0.61 |
| Unclassified PH | PVR $\leq$ 2, PAWP $\leq$ 15 | 3 | 7.04 | 0.94 |
| CpcPH | PVR $>$ 2, PAWP $>$ 15 | 27 | 8.88 | 0.50 |
| IpcPH | PVR $\leq$ 2, PAWP $>$ 15 | 4 | 7.18 | 0.22 |

