# OpenReview forum: "Non-invasive estimation of haemodynamic parameters in pulmonary hypertension — A deep learning approach integrating all B-mode cine loops in an echocardiographic exam"
_MIDL.io/2026/Conference — MIDL 2026 Poster_

### Official Review · Reviewer_bEDZ · 2026-01-11

**Confidence:** 4
**Preliminary Rating:** 4

**Summary:**

This paper investigates the feasibility of estimating key haemodynamic parameters in pulmonary hypertension (PH)—including mean pulmonary artery pressure (mPAP), pulmonary vascular resistance (PVR), and pulmonary arterial wedge pressure (PAWP)—non-invasively using deep learning applied to echocardiographic B-mode cine loops. The proposed framework integrates all available cine loops from an exam, without requiring view annotations, using a multi-stage attention-based feature fusion architecture.

**Strengths:**

Methodologically, the ability to ingest an arbitrary number of view-unannotated cine loops is a notable design choice that aligns well with real-world echocardiographic workflows, where view labels are often missing or unreliable. The attention-based fusion mechanism is well justified and provides a degree of interpretability by highlighting clinically relevant views. The paper also goes beyond single-parameter prediction by estimating a full haemodynamic profile (mPAP, PVR, PAWP), which is clinically meaningful.

The manuscript is clearly written, well structured, and situates the work appropriately within prior literature. Experimental protocols are transparent, and the discussion acknowledges both strengths and limitations in a balanced manner.

**Weaknesses:**

While the proposed framework is well designed and evaluated, several aspects could be further strengthened. First, although mPAP prediction shows moderate correlation with RHC, specificity remains low, with a tendency to overestimate elevated pressures. Additional analysis of calibration, error behavior around clinical thresholds, or uncertainty-aware prediction would help clarify appropriate clinical use, particularly for ruling out PH.

Second, the attention analysis is largely qualitative and based on a limited subset of highly weighted cine loops. A more systematic, cohort-level assessment of view-wise contribution stability would strengthen confidence in the robustness of the learned attention mechanism.

Third, the echocardiography–RHC interval can extend to three months, potentially introducing label noise due to disease progression or treatment effects. A sensitivity analysis stratified by time interval would help quantify its impact on model performance.

Finally, the evaluation is cross-sectional. Discussion or preliminary analysis of longitudinal consistency, such as test–retest stability or tracking haemodynamic changes over time, would further enhance the clinical relevance of the approach.

**Detailed Comments:**

Reporting calibration curves or Bland–Altman plots for mPAP would improve clinical interpretability.

Clarifying whether performance differs across PH subgroups (e.g., pre-capillary vs post-capillary) would be valuable.

The pre-training strategy for the frame encoder is well motivated; brief ablation results would further support its contribution.

**Justification Of The Preliminary Rating:**

The ability to integrate all available cine loops without view annotations is a practical strength, and the results suggest meaningful potential as a screening or assistive tool rather than a replacement for RHC. Overall, the work is solid, clearly presented, and of interest to the medical imaging and clinical AI community, supporting a weak accept decision.

**Questions To Address In The Rebuttal:**

How should the model outputs be interpreted clinically given the low specificity for normal mPAP values?

How sensitive are the results to the echocardiography–RHC time interval?

Does the attention mechanism select consistent views across patients, or is it highly exam-specific?

Could the framework be extended to longitudinal monitoring of PH progression or treatment response?

---

> ### Author Response · Authors · 2026-01-24
>
> Thank you for the positive comments on the model’s attention fusion mechanism, and its ability to predict a full haemodynamic profile. Below we discuss each raised weakness (W), question (Q), and detailed comment (DC).
>
> * **W1.Q1: Low specificity**
>
>   To answer this question comprehensively, we posted *Comment A* about model performance and class balancing.
>
> * **W2.Q3: Attention analysis**
>
>   Thank you for pointing out the question. The limitation links to the background that this work was built on: annotations of view and image quality are usually not a given in real world datasets. Effort was made to summarize the characteristics of relevant and irrelevant cine loops quantitatively, e.g. 12 A4C views out of 16 most weighted cine loops; 11 suboptimal image quality out of 16 least preferred cine loops. However, we acknowledge that we do not have the resources to annotate the entire test set for view and image quality to extend such summarization to the entire test set. As a work around, we focused the annotation effort to the most and the least weighted cine loops, attempting to still conclude what are the representative characteristics for relevant and irrelevant cine loops, respectively. With the available annotations, we currently cannot yet ensure whether the most-weighted view across the exams were all A4C, knowing only the most-weighted cine loops are frequently A4C. Ideally, the entire test set should be annotated, and the averaged attention weight per view and per image quality could be derived. We therefore included the following in the last paragraph in the Discussion:
>   > *Finally, model interpretability remains limited. If view annotations become available in the complete testing set, analyzing individual view contributions could refine our understanding of their importance.*
>
> * **W3.Q2: Echo–RHC time interval**
>
>   To answer this question comprehensively, we posted *Comment B* about the time gap between echo and RHC.
>
> * **W4.Q4: Track change**
>
>   We agree with the reviewer that the potential of the model to track changes over time or across treatment would be an important aspect to verify. Currently we have only 1 patient in the test set with 2 exams, which is unfortunately not yet sufficient to support such evaluation.
>
> * **DC1. DC2: Detailed analysis on mPAP prediction**
>
>   Thank you for the suggestions to help understanding the model behavior better. We now added the Bland–Altman plot for mPAP prediction in Appendix D (Figure 6). The plot showed positive mean bias (≈ 1.3 mmHg), and a wide limits of agreement (-20.0 to 22.6 mmHg), indicating substantial variability at the individual level. The comparison of predictive performance across different subgroups was also added to Appendix D (Table 6). It indicated that performance in PH subgroups was still dominant by the majority, being the pre-capillary group. The isolated post-capillary PH group had especially low PCC, but the group was also under represented in the dataset. However, the largest error was still observed in non-PH subjects, which remains the major point to improve in future works.
>
> * **DC3: Pre-training**
>
>   Pre-training was indeed found to be an important method to familiarize the Frame Encoder with the diversity in echo images, including different views, zoom levels, and focal points. Pre-training improved the predictive performance as seen in Table 5 (a, b) in the added Appendix C (MAE: 10.05→8.74, PCC: 0.66→0.73). We adopted a basic image reconstruction task to pre-train the encoder, but different ways to initialize the encoder are definitely worth exploring in future works. For example, masked autoencoder, DINO, contrastive learning, or adopting a relevant foundation model.

---

### Official Review · Reviewer_SBCU · 2026-01-13

**Confidence:** 4
**Preliminary Rating:** 5
**Final Rating:** 5

**Summary:**

This paper explores the feasibility of non-invasively estimating haemodynamic parameters for Pulmonary Hypertension using a deep learning approach that integrates all B-mode echocardiographic cine loops from a single exam. The significance of this work lies in its innovative model architecture, which utilizes an automated, data-driven attention mechanism to objectively aggregate comprehensive information from diverse, unannotated cine loops.

**Strengths:**

1.  The proposed architecture is significantly valuable due to its view-agnostic nature, allowing it to process an arbitrary number of echocardiographic cine loops without requiring manual view annotations

2. A major strength lies in the model's reliance solely on B-mode images, enabling it to provide haemodynamic estimates for the 24% of patients who lack a measurable tricuspid regurgitation TRV_max signal

3.  The paper is well-structured, follows sound scientific principles, and adequately addresses prior work. The inclusion of a pre-training phase for the Frame Encoder using an image reconstruction task is a clever way to familiarize the model with the diversity of ultrasound images before the supervised learning task.

**Weaknesses:**

1. The use of the ASPIRE database introduces an inherent high-prevalence bias, yet the methodology lacks specific strategies—such as loss weighting or balanced sampling—to address this skewness. This absence is reflected in the low specificity of 0.24, suggesting that the model significantly struggles to accurately exclude non-PH cases in a clinical screening context. Integrating data-balancing techniques could potentially reduce these false positives and enhance the model's discriminative utility.

2. While the model predicts mPAP, PVR, and PAWP using a single architecture, the notably lower PCC of 0.54 for PAWP highlights a methodological bottleneck. Given that mPAP relates to right ventricular morphology and PAWP to left heart filling pressure, these parameters arise from distinct physiological mechanisms. A "one-size-fits-all" encoder may be insufficient; a specialized feature fusion strategy for PAWP could better capture these disparate signals.

3. The 3-month inclusion window allows for significant haemodynamic fluctuations, yet the study does not detail any filtering for clinical stability, such as changes in WHO functional class or medication. This temporal discrepancy introduces label noise, which likely contributes to the model's difficulty in predicting extreme values and its tendency toward "regression to the mean". Clarifying the correlation between the time interval and prediction error would help validate the model's robustness against such dynamic physiological changes.

**Detailed Comments:**

The paper's methodology mentions uniformly scaling and cropping all images to 256x256. However, for cardiac ultrasound, many subtle structural movements (such as subtle signs of ventricular septal flattening) may be lost in this low-resolution processing. Have the authors considered spatial pyramid pooling or other methods that preserve high-resolution features?

**Justification Of Final Rating:**

The authors have adequately addressed the peer review comments.
However, the time gap between echocardiography and right heart catheterization (RHC) remains a limitation of the dataset.
My overall review rating therefore remains unchanged.

**Justification Of The Preliminary Rating:**

This paper presents an innovative model architecture and addresses a clinically significant problem. While these weaknesses require further clarification, the paper’s contribution to automated, view-agnostic echocardiographic analysis remains highly valuable to the community.

**Questions To Address In The Rebuttal:**

1. Did the authors investigate the use of weighted loss functions or balanced mini-batch sampling during the training of the multi-view model? How might these technical adjustments influence the model's ability to accurately classify non-PH cases?

2. Did the authors perform any sensitivity analysis regarding the time-gap between echocardiography and RHC? Specifically, does the prediction error correlate with the length of this interval, and was there any filtering based on clinical stability (e.g., medication or WHO class changes)?

---

> ### Author Response · Authors · 2026-01-24
>
> Thank you for the compliments on the benefits resulting from the model architecture, and the positive feedback in general. Below we discuss each raised weakness (W), question (Q), and detailed comment (DC).
> * **W1.Q1: Class balancing**
>
>   To answer this question comprehensively, we posted *Comment A* about model performance and class balancing.
>
> * **W2: Specialized model for each target**
>
>   We appreciate the insightful comments. Although mPAP and PAWP arise from distinct physiological mechanisms, the underlying cardiac structures are anatomically and functionally inter-related. From this perspective, sharing an image encoder across different targets can still be beneficial, while allowing for specialized downstream regressors. In this design, the encoder focuses solely on extracting the underlying semantic features from the images, which vary in views, zoom levels, and focal points. The entire framework can stay lightweight, and scales well to more prediction targets or more input views.
>
>   Within the current framework, predictive performance for individual haemodynamic parameters could potentially be improved, by ensuring that the most informative view(s) for each target are at least present in the input ensemble, and are of adequate image quality. This can be done manually or with specialized models. Our proposed framework provides a work-around for the current limitations in view and image quality annotations, but incorporates well with the view selection and image quality assessment models as they become mature in the future.
>
> * **W3.Q2: Echo–RHC time gap**
>
>   To answer this question comprehensively, we posted *Comment B* about the time gap between echo and RHC.
>
> * **DC1: Preserving resolution**
>
>   Thank you for the constructive suggestion. In this work, we followed a common image preprocessing workflow, and ensured the content in the imaging sector was preserved in the cropped image. However, preserving both spatial and temporal resolution for the crucial region and phase will indeed be an important direction to explore.

---

### Official Review · Reviewer_zXaU · 2026-01-15

**Confidence:** 3
**Preliminary Rating:** 3
**Final Rating:** 3

**Summary:**

This paper presents a deep learning framework for estimation of haemodynamic parameters using echocardiographic B-mode cine loops. The framework is designed to accept an arbitrary number of view-unannotated loops per study and uses attention-based feature fusion, both temporally and spatially, to integrate the input information.
A large retrospective dataset is used to evaluate the approach. The results are reported in terms of correlation with ground truth, classification accuracy (PH vs. non-PH) and in terms of clinically relevant views, as suggested by the attention weights of the deep learning model.

**Strengths:**

* A non-invasive way of estimating haemodynamic metrics has high clinical relevance.
* The method is evaluated on a large, real-world dataset.
* The ability to process an arbitrary number of cine loops is novel and addresses real-world data constraints.
* The paper tries to interpret the attention weights together with expert annotations to identify clinically relevant and irrelevant cine loops/views.
* Strong clinical study

**Weaknesses:**

* Limited predictive performance: While the authors acknowledge and discuss that, the specificity is poor, raising concerns about the clinical applicability of the method.
* The architecture is largely an adaptation of prior work from the authors
* Incremental ML contribution
* Source code does not seem to be public

**Detailed Comments:**

* Minor typo in §2.5: "Pytorch and Lightning" -> "Pytorch and Pytorch Lightning". And two sentences later: "with early stop" -> "with early stopping".

**Justification Of Final Rating:**

The authors have addressed my questions regarding data splitting, loss-functions, and class imbalance, and have clearly acknowledged the limitations. While the rebuttal improves confidence in the evaluation and transparency of the study, the overall ML contribution remains incremental, and my preliminary score remains unchanged.
The paper is better suited for a clinical or validation-focused track rather than a technical ML conference due to its clinical importance but limited technical innovation and unresolved predictive performance issues.

**Justification Of The Preliminary Rating:**

This paper appears to be a strong clinical study with only incremental ML contribution. It addresses a real clinical problem and shows a feasible solution to that. However, the architecture seems to be an incremental adaptation of prior work, extended with well-established techniques. The technical depth of the paper is somewhat limited and mainly focused on clinical interpretation. The paper could be tailored towards a more technical audience by, e.g., explicitly analyzing the method's capabilities as a set-function learner, doing ablation studies, or presenting novel methodology. Maybe this would be better suited for the special track on validation studies.

**Questions To Address In The Rebuttal:**

* Please clarify whether the random train/validation/test split was done at the patient level, with no subject overlap between the sets.
* Ablation study on loss functions: Did the additional BCE loss improve regression accuracy, and vice versa?
* How many cine loops were on average used per case during testing?

---

> ### Author Response · Authors · 2026-01-24
>
> Thank you for the positive comments on the clinical application aspect of the work. Below we discuss each raised weakness (W), question (Q), and detailed comment (DC).
> * **W1: Limited predictive performance**
>
>   To answer this question comprehensively, we posted *Comment A* about model performance and class balancing.
>
> * **W2.W3: Adaptation of prior work and incremental ML contribution**
>
>   With limited data size as compared to the computer vision field, we intentionally start always by investigating previously verified models to avoid overfitting the design to a specific dataset. However, effort was made in adapting our previously developed framework to the current work.
>   * Stage 1 view-wise fusion in the prior work had to be removed for unsynchronized cine loops.
>   * Pre-training was found to be a key element to make the model perform well on the ultrasound dataset, which has substantial variability in imaging views, zoom levels, and focal points. (Table5 (a, b) in Appendix C)
>   * BCE was added to the loss function to improve delineation around the clinical threshold. (Table5 (a, c) in Appendix C)
>
>   Additional potentials of the previously proposed framework are now verified, which was not possible before and were listed only as discussions.
>   * The framework is now shown to be applicable on B-mode ultrasound cine loops as well, which are much less standardized and thus more challenging, compared to MR cine series in the prior work. The attention mechanism was also found to be helpful for down-weighting the low-quality images.
>   * Integrating more views is now shown to be possible, as opposed to integrating 4 views in the prior work.
>   * Integrating unannotated cine loops is now shown to be possible, which no longer confines image diversity within only pre-defined views, and the input exams are no longer guaranteed to contain no missing or repeated views.
>
> * **W4: Code availability**
>
>   The code and the trained models can be shared upon reasonable request.
>
> * **Q1: Data splits**
>
>   We confirm that the random train/validation/test split was done at the patient level, to make sure exams from the same patient will not leak into different splits.
>
> * **Q2: Loss function ablations**
>
>   From the added Table 5 (a, c) in Appendix C, we observed that adding BCE loss did improved the regression performance as well (MAE: 9.31→8.74, PCC: 0.72→0.73). We did not train the model with only BCE loss, which would have enabled a comparison in the opposite direction. We value that the framework predicts a continuous haemodynamic profile, but training with only BCE loss would indeed be an informative ablation. We thank the reviewer for the insightful question.
>
> * **Q3: Number of cine loops per case during testing**
>
>   In testing, all B-mode cine loops in an exam were used as the model input. The average is 23.6 cine loops per exam. Additionally, the median is 23, and the standard deviation is 8.2.
>
> * **DC1: Typo**
>
>   The typo in Section 2.5 was fixed.

---

### Author Rebuttal · Authors · 2026-01-24

**Rebuttal:**

## **Summary of manuscript modifications:**
* **New sections added to the end of the Appendix:**

  "*Appendix C. Ablation study*", and "*Appendix D. mPAP prediction details*", with new figure (Figure 6) and tables (Table 5, 6).

* **Typo fixed in section 2.5:**

  "Pytorch Lightning"; "early stopping"

* **Minor text updates in the last paragraph of section 4:**

  concerns "the" predictive performance; in the "complete" testing set

(All modifications were highlighted in orange in the updated manuscript.)

**Supporting Material:**

/attachment/075b33c97388dd75c411e8c94868699e46840d92.pdf

---

### Author Response · Authors · 2026-01-24

### **Comment A**: Model performance and class balancing
Apart from the MSE loss, the BCE loss was added initially for the exact reason to improve the classification performance, under the framework that the model still predicts a continuous haemodynamic profile. We now added the details of the ablation study in Appendix C. The ablation study showed that adding BCE loss improved the specificity from 0.19 (Table5(c)) to 0.30 (Table5(a)).
However, when using weighted BCE loss (Table5(d), specificity=0.27), or weighted sampling (Table5(e), specificity=0.30), there was no further improvement in specificity. The former down-weights the loss for the majority class. The latter oversampled the minority-class exams during mini-batch formation.

The proposed model demonstrates high sensitivity but lower specificity, reflecting the relative underrepresentation of normal exams in the training data. In a referral-based population, such as patients with suspected pulmonary hypertension evaluated at a specialized center, this performance profile (sensitivity at testing = 0.94) can be acceptable, as the model prioritizes identification of patients with elevated mPAP while minimizing missed PH cases. In this context, false-positive predictions may prompt further diagnostic evaluation but are unlikely to result in harmful downstream consequences. However, for use in a screening setting or in populations with a higher percentage of normal mPAP, the current specificity (0.24 at testing) would limit clinical applicability. In such scenarios, fine-tuning using data with a representative mPAP distribution would be required to improve specificity. This was therefore acknowledged in the last paragraph of the Discussion:
> *A major limitation of the current work concerns the predictive performance. Particularly, the specificity of the mPAP model remains to be improved. The high false positive rate likely arises from dataset imbalance, as the cohort contains more patients with high mPAP than those with normal values. Addressing this may require expanding the dataset to include more non-PH cases or employing data-balancing techniques.*
---
### **Comment B**: Time gap between echo and RHC
The initial reason we allowed a time gap between echo and RHC was to maintain a large data size, and avoid excluding exams that could still be valuable for training, despite potentially with a noisy label. In the current dataset, diagnosis was based on the RHC results. Unfortunately, we do not have further details on changes in WHO functional class or medication between echo and RHC. It is also uncertain if the patients with a long time interval between echo and RHC came from a certain subgroup.

The additional plot in the link below presents absolute difference between the predicted and RHC-measured mPAP, against the time gap. A positive time difference implies RHC was performed later than echo. The scatter plot reveals that the magnitude of the error is not associated with the time difference. If the correlation trend was clear, it would probably make sense to sacrifice data size and exclude exams with a larger time gap. However, we acknowledge that, if possible, a smaller time gap between RHC and echo, guaranteeing no changes in medication and WHO functional class, would still be more ideal. We therefore included the following in the last paragraph of the Discussion:
> *Another limitation arises from the potential variations in patient status between the RHC and echocardiography. It would be ideal if the two tests would be performed closer to each other, to further reduce potential label noise.*

**Link to plot:**
https://github.com/LishinC/SPHDUSrebuttal/blob/ccf628f02187bee95c9c314039c46c5a8cd95ab2/rebuttal_time_gap.png?raw=true

---

### Meta-Review · Area_Chair_uCu5 · 2026-02-09

**Recommendation:** Accept (Poster)
**Confidence:** 5

**Metareview:**

This paper received strong accept, weak accept, and borderline.

The final decision reflects the reviewers’ overall evaluations. The authors are encouraged to carefully address the remaining concerns and incorporate key clarifications from the rebuttal into the final manuscript.

The primary outstanding concern is that the overall machine learning contribution remains incremental; however, the reviewer acknowledges the paper’s clinical importance.

---

### Decision · Program_Chairs · 2026-02-13

Accept (Poster)